# Sustained viremia suppression by SHIV$_{SF162P3CN}$-recalled effector-memory CD8$^+$ T cells after PD1-based vaccination

Yik Chun Wong[1,2☯], Wan Liu[1☯], Lok Yan Yim[1,2☯], Xin Li[1,3☯], Hui Wang[2☯], Ming Yue[4☯], Mengyue Niu[1], Lin Cheng[2], Lijun Ling[1], Yanhua Du[1], Samantha M. Y. Chen[1], Ka-Wai Cheung[1], Haibo Wang[1], Xian Tang[2,5], Jiansong Tang[1], Haoji Zhang[3], Youqiang Song[4], Lisa A. Chakrabarti[5], Zhiwei Chen[1,2]*

**1** AIDS Institute, Department of Microbiology, State Key Laboratory of Emerging Infectious Diseases, Li Ka Shing Faculty of Medicine, University of Hong Kong, Hong Kong SAR, China, **2** HKU-AIDS Institute Shenzhen Research Laboratory and AIDS Clinical Research Laboratory, Guangdong Key Laboratory of Emerging Infectious Diseases, Shenzhen Key Laboratory of Infection and Immunity, Shenzhen Third People's Hospital, Shenzhen, China, **3** Department of Veterinary Medicine, Foshan University, Foshan, China, **4** School of Biomedical Sciences, Li Ka Shing Faculty of Medicine, University of Hong Kong, Hong Kong SAR, China, **5** Virus and Immunity Unit, Pasteur Institute, Paris, France; INSERM U1108, Paris, France

☯ These authors contributed equally to this work.
* zchenai@hku.hk

**Data Availability Statement:** All single-cell RNA sequencing raw data files are available from the NCBI's Gene Expression Omnibus (GEO) and are accessible through GEO Series accession number

## Abstract

HIV-1 functional cure requires sustained viral suppression without antiretroviral therapy. While effector-memory CD8$^+$ T lymphocytes are essential for viremia control, few vaccines elicit such cellular immunity that could be potently recalled upon viral infection. Here, we investigated a program death-1 (PD1)-based vaccine by fusion of simian immunodeficiency virus capsid antigen to soluble PD1. Homologous vaccinations suppressed setpoint viremia to undetectable levels in vaccinated macaques following a high-dose intravenous challenge by the pathogenic SHIV$_{SF162P3CN}$. Poly-functional effector-memory CD8$^+$ T cells were not only induced after vaccination, but were also recalled upon viral challenge for viremia control as determined by CD8 depletion. Vaccine-induced effector memory CD8$^+$ subsets displayed high cytotoxicity-related genes by single-cell analysis. Vaccinees with sustained viremia suppression for over two years responded to boost vaccination without viral rebound. These results demonstrated that PD1-based vaccine-induced effector-memory CD8$^+$ T cells were recalled by AIDS virus infection, providing a potential immunotherapy for functional cure.

## Author summary

HIV-1/AIDS remains a major global pandemic although treatment regimen has improved. Identifying efficacious vaccines and therapeutics to achieve long-term viral control with very low/undetectable plasma viral loads in the absence of antiretroviral therapy, a status known as functional cure, would be highly beneficial. We previously demonstrated that antigens fused to a soluble program death-1 (PD1) domain could effectively bind and be cross-presented by dendritic cells that constitutively expressed PD1 ligands.

GSE171783 via the link (https://www.ncbi.nlm.nih.gov/geo/query/acc.cgi?acc=GSE171783).

**Funding:** This work was supported by the Hong Kong Research Grant Council (RGC) via Theme-based Research Scheme (T11-706/18-N), The French National Research Agency (Agency Nationale de la Recherche)/RGC Joint Research Scheme (A-HKU709/14), General Research Fund (762712) and Collaborative Research Fund (HKU5/CRF/13G). This work was also supported by Health and Medical Research Fund (17160762) from Food and Health Bureau, The Government of Hong Kong Special Administration Region of the People?s Republic of China, University Development Fund (http://www.rss.hku.hk/news/page/2) from The University of Hong Kong, Li Ka Shing Faculty of Medicine Matching Fund (http://www.rss.hku.hk/news/page/2) from The University of Hong Kong, Mega-Projects of National Science Research for the 13th Five-Year Plan in China (2018ZX10731101-002-001) and the Sanming Project of Medicine in Shenzhen (AR160087). The funders had no role in study design, data collection and analysis, decision to publish, or preparation of the manuscript.

**Competing interests:** I have read the journal's policy and the authors of this manuscript have the following competing interests: Z.C. is a co-inventor on the PD1-based vaccine patent. However, ZC and all other authors do not have any competing interests or additional financial interests.

When applied in the form of DNA vaccination, this antigen-targeting strategy was highly immunogenic in mice. Here, we investigated the efficacy of the PD1-based DNA vaccine approach against pathogenic simian-human immunodeficiency virus challenge in rhesus monkeys. Our results showed that homologous PD1-based DNA vaccinations induced highly functional effector-memory CD8+ T cells carrying a unique cytotoxicity gene expression profile. These T cells actively supressed viremia in monkeys and were re-activated via boost vaccination at 2 years after viral challenge without viral rebound. In summary, our study demonstrates the potential application of PD1-based DNA vaccination to control AIDS virus infection.

## Introduction

Life-long treatment with combination antiretroviral therapy (cART) is required to supress viral replication in human immunodeficiency virus-1 (HIV-1) infected patients because cART does not completely eliminate the virus. Functional cure, as defined by sustained viremia suppression without cART [1], represents an alternative strategy to combat the HIV-1/AIDS epidemic. Previous studies demonstrated the essential role of T cells, in particular effector-memory CD8+ T cells, in mediating viral suppression in HIV-1 patients [2–5] and in rhesus macaques infected with simian immunodeficiency virus (SIV) [6,7]. Induction of broad and high frequency of anti-viral cytotoxic CD8+ T cell response represents a favourable immunotherapeutic approach for sustained HIV-1 viremia control.

Many T cell vaccine strategies, for example, the use of novel cytomegalovirus (CMV)-, vaccinia- or adenovirus-based vectors [8–11], prime-boost regimens of various vaccine combinations [12–14], or unique immunogen designs such as mosaic antigens to cover diverse pathogen variants [15–19], elicit protective T cell immunity against SIV or simian-human immunodeficiency virus (SHIV) infections in preclinical macaque studies [14]. However, investigation on the mechanisms underlying sustained setpoint viremia suppression by vaccination alone remains incomplete in non-human primate (NHP) models.

To improve the potency of AIDS vaccines, several methods of targeting HIV-1 antigen to dendritic cells (DC) have been tested in NHP by others [20–22]. One approach was fusion of antigens to an antibody that targets DC-specific surface molecules, such as DEC205 endocytosis receptor [23]. This method, however, mainly induced antigen-specific CD4+ T cells [20–22], the preferential targets of HIV-1 infection. In comparison, a different DC-targeting vaccine strategy to improve CD8+ T cell immunity has been established. This strategy involves fusing HIV-1 antigen to a soluble program death-1 (PD1) domain (sPD1) [24]. PD1 ligands are constitutively expressed on DC, and antigens tagged with sPD1 are preferentially targeted to DC for cross-presentation to CD8+ T cells. Compared to non-DC targeting controls, CD8+ T cells induced by this vaccine strategy provided significantly improved protection against poxvirus infection and tumour growth in mice [24,25]. However, the potency of PD1-based vaccine remains elusive in NHP, the closest animal model of HIV-1 infection.

In this study, we assessed the immunogenicity and protective efficacy of the PD1-based vaccine approach in mice and subsequently brought the promising candidate into NHP against high-dose intravenous challenge by the CCR5-tropic, neutralization tier-2 pathogenic SHIV$_{SF162P3CN}$ in rhesus macaques of Chinese origin. We demonstrated that PD1-based DNA vaccination induced poly-functional effector-memory CD8+ T cells that was potently recalled upon SHIV$_{SF162P3CN}$ infection to sustain long-term viremia suppression.

## Results

### A rhesus PD1-based vaccine encoding the SIV capsid antigen induced stronger T cell immune responses and protection in mice

Better HIV-1 control is associated with strong CD8$^+$ T cell responses against the capsid antigen, but not other Gag regions [26,27]. We previously demonstrated that murine PD1-based DNA vaccine msPD1-p24 was much better than p24 alone especially for inducing stronger antigen-specific CD8$^+$ T cells through antigen cross presentation [24]. Taking one step forward, we generated a similar monkey vaccine, namely pRhPD1-p27, to express a rhesus sPD1-fused SIV Gag-p27 capsid antigen to compare with a pair of DNA vaccines encoding a full-length SIV Gag antigen, either alone (pGag) or fused to rhesus sPD1 (pRhPD1-Gag; S1A Fig). The capacity of rhesus sPD1-tagged antigens to bind to mouse PD-L1 and PD-L2 was confirmed by flow cytometry (S1B Fig).

To compare the immunogenicity of these constructs, C57BL/6 mice were immunised 3 times at 3-week intervals with 100 μg of each DNA vaccine, using intramuscular injection with electroporation (im/EP; Fig 1A). The pRhPD1-p27 vaccine induced the highest amounts of Gag-specific IgG antibody response by ELISA (Fig 1B), T cell responses against the Gag antigen by interferon-γ (IFN-γ) ELISpot (Fig 1C) and the immunodominant AL11 epitope by the H-2D$^b$/AL11 tetramer assay (Fig 1D and 1E) as compared with other DNA vaccines. Importantly, the pRhPD1-p27 vaccination also conferred a better protection against challenge with a replication-competent modified vaccinia virus Tiantan, namely MVTT$_{SIVgpe}$ [12], which expressed SIV Gag, Pol and Env antigens (SIV$_{gpe}$), as compared to pRhPD1-Gag (Fig 1F). Moreover, the Gag-specific memory antibody and CD8$^+$ T cell responses were at least 2-fold higher in pRhPD1-p27-vaccinated mice than those immunised with pRhPD1-Gag or pGag (Fig 1G–1I). Our results demonstrated that pRhPD1-p27 has better immunogenicity, and therefore was selected for subsequent study in NHP.

### pRhPD1-p27 vaccination conferred protection against high-dose challenge using the pathogenic SHIV$_{SF162P3CN}$ in rhesus macaques

To determine the potential of the pRhPD1-p27 DNA vaccine in NHP, we have first passaged the pathogenic SHIV$_{SF162P3}$ strain, which was isolated from an Indian macaque with simian AIDS [28,29], in peripheral blood mononuclear cells (PBMC) of Chinese macaques to avoid immune response to cellular antigens derived from Indian macaques [30]. The adapted strain was named SHIV$_{SF162P3CN}$, a neutralization-resistant tier-2 virus that uses CCR5 as the only coreceptor similar to its parental strain [28,29]. We then examined the vaccine in two experimental groups consecutively. The group A involved a longer vaccination schedule than the subsequently repeated group B (Fig 2A). Chinese-origin rhesus macaques were vaccinated via im/EP with pRhPD1-p27 four times at different intervals. Control macaques were not immunized (unvaccinated). Considering the small number of animals in each group, we challenged individual macaque with a high stringent dose (5000 TCID$_{50}$) of SHIV$_{SF162P3CN}$ intravenously at least 20 weeks after the last immunisation with focus on recalled immune responses for protection.

Upon viral challenge, all macaques showed infection with measurable viral loads (Fig 2B). The plasma viral RNA loads (pVL) of control animals from both experiments (n = 9) peaked at 1.5–2 weeks post infection (wpi), with 9.95x10$^6$ copies/ml in average. During the setpoint phase at 6–20 wpi, these macaques had a geometric mean pVL of 4.58x10$^4$ copies/ml. This demonstrated the high pathogenicity of intravenous challenge using a large dose of SHIV$_{SF162P3CN}$. In comparison, although the 4 vaccinated macaques in group A had a similar peak pVL as those of controls, their pVL were subsequently reduced to undetectable levels

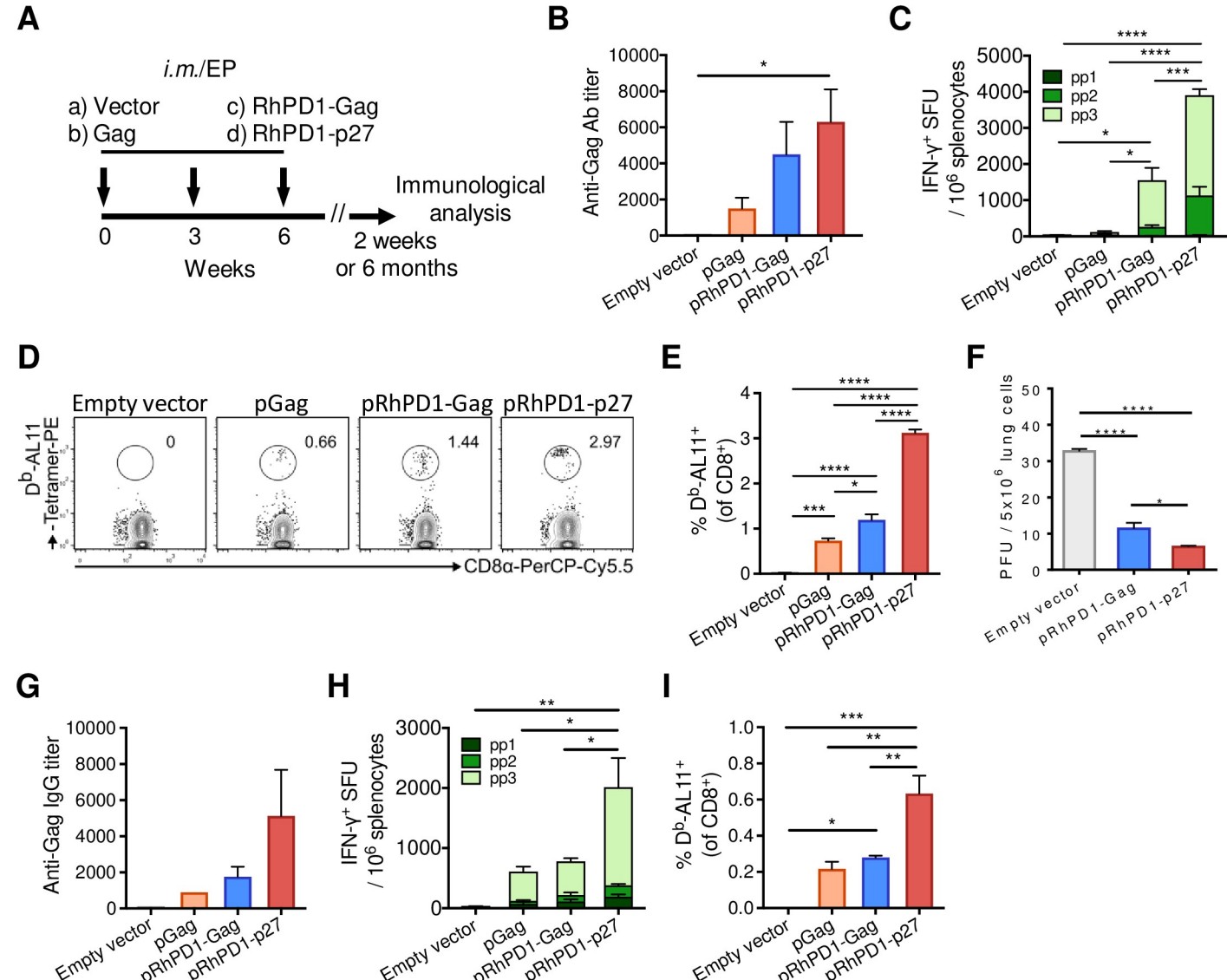

**Fig 1. Immunogenicity of RhPD1-based vaccines in mice.** (**A**) Schematic of DNA vaccination in mice. C57BL/6 mice were immunized with the empty pVAX vector, pGag, pRhPD1-Gag or pRhPD1-p27 via intramuscular injection with 3 consecutive electroporation in 3-week intervals. Immunological analysis was performed at 2 weeks or 6 months after the last immunization. (**B**)—(**E**) Acute humoral and cellular immunity induced by sPD1-based vaccines 2 weeks post-last immunization. (**B**) Anti-Gag antibody titers in sera. (**C**) Gag-specific cellular immune responses in spleens evaluated by IFN-γ ELISpot assays. (**D**) Representative FACS plots showing the presence of H-2D[b]/AL11-specific CD8[+] T cells in the spleens of vaccinated mice by tetramer staining analysis. (**E**) Frequencies of H-2D[b]/ AL11-specific CD8[+] T cells in vaccinated mice. (**F**) The efficacy of the sPD1-based vaccines against recombinant vaccinia virus challenge. Two weeks after last immunization, mice were challenged with recombinant vaccinia virus MVTT-SIV$_{gpe}$ via intranasal route. Viral loads in the lungs were determined two days after challenge. (**G**)—(**I**) Memory humoral and cellular immune responses elicited by the sPD1-based vaccines at 6 months post last immunization. (**G**) Anti-Gag IgG titers in sera. (**H**) Gag-specific IFN-γ ELISpot responses and (**I**) the frequencies of H-2D[b]/AL11-specific CD8[+] T cells in spleens.

during the setpoint phase (Fig 2B). In group B that had a shortened vaccination schedule, the pVL peaked at $9.26 \times 10^4$ copies/ml, and gradually decreased to below the detection limit. Comparing the two vaccinated groups, macaques in group B had a lower peak viremia than those in group A by 3 $\log_{10}$ (Fig 2C), likely suggesting a better efficacy of the shortened vaccination protocol. Since the vaccinated macaques in both groups A and B had a similarly low setpoint pVL, the setpoint values were combined. Compared to controls, the pRhPD1-p27 vaccine significantly reduced setpoint viremia (Fig 2D) and proviral DNA loads at 12–13 weeks post-challenge

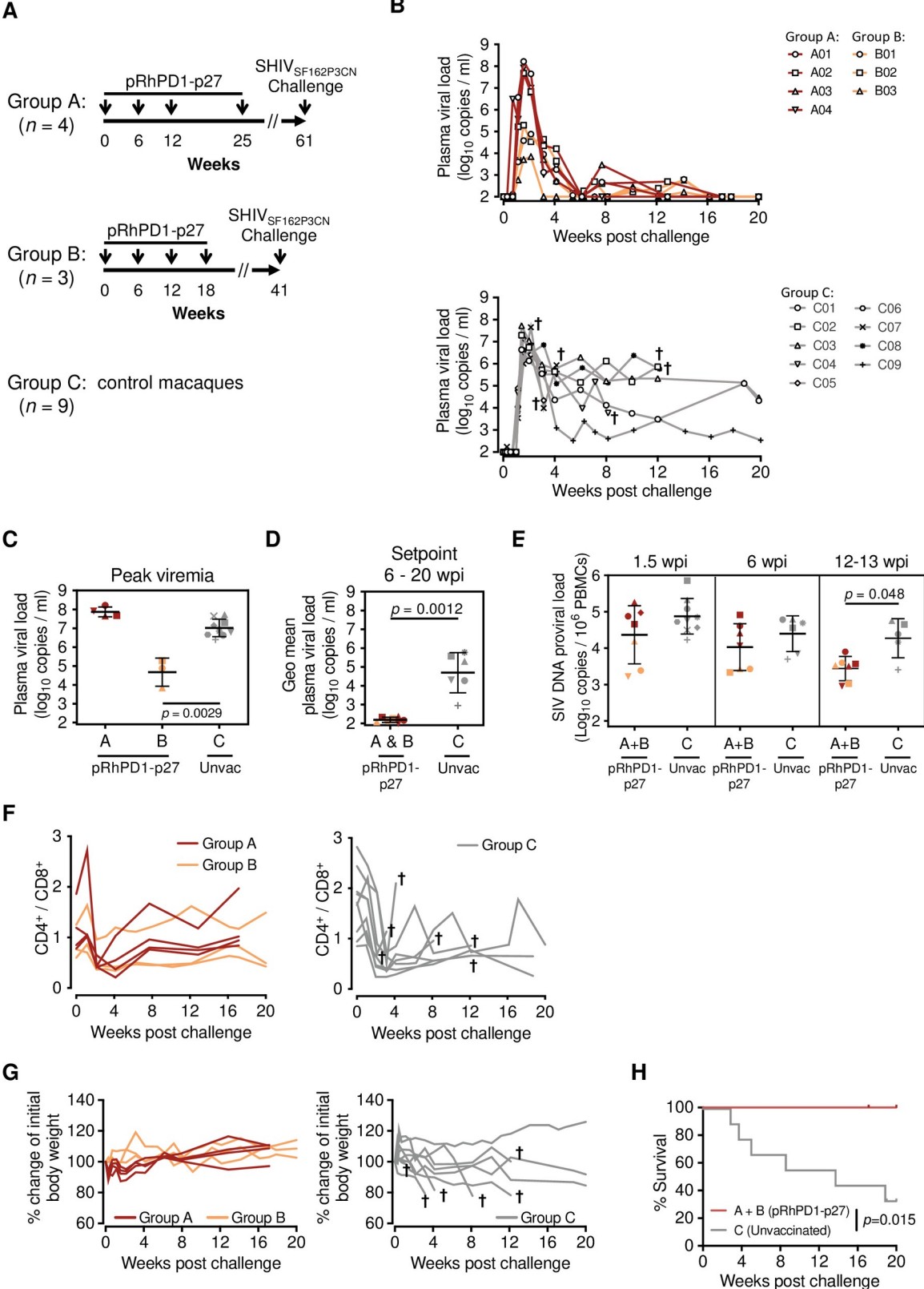

**Fig 2. Viral suppression in rhesus macaques immunised with the PD1-based pRhPD1-p27 DNA vaccine after high dose pathogenic SHIV_{SF162P3CN} challenge. (A)** Schematic of DNA vaccination in Chinese-origin rhesus macaques. Two studies with different

vaccination durations were conducted. In both studies, Chinese-origin rhesus macaques were immunised with the PD1-based pRhPD1-p27 DNA vaccine via intramuscular injection with electroporation 4 times in 6- to 13-week intervals, followed by intravenous challenge with 5000 $TCID_{50}$ of $SHIV_{SF162P3CN}$ at 23 or 36 weeks after last immunization. Unvaccinated animals were included as controls. **(B)** Plasma viral loads of the vaccinated (top) and unvaccinated macaques (below) after SHIV challenge. Animals that were euthanised were marked with †. Anti-CD8β antibody (clone CD8b255R1) was infused into the vaccinated macaques of Group A intravenously at 17 weeks post-challenge, after confirming pVL were below detection levels. Data of this treatment is presented in Fig 5D. **(C)** Comparison of the peak viremia and **(D)** geometric means of setpoint viremia determined from 6–20 weeks post-infection (wpi). **(E)** Proviral DNA loads in PBMC in vaccinated and unvaccinated macaques. Geometric means ± geometric SD are shown in (C)–(E). Significances of differences were determined by the Kruskal-Wallis test followed by the Dunn's multiple comparisons test (C), or the two-tailed Wilcoxon rank sum test (D and E). **(F)** The CD4+/CD8+ T cell ratios in vaccinated rhesus macaques from Group A and B (left) or unvaccinated macaques (right) after $SHIV_{SF162P3CN}$ challenge. **(G)** Changes of body weight of the vaccinated (left) and unvaccinated macaques (right). Animals that were euthanised over the measurement period were marked with †. **(H)** Survival of vaccinated and unvaccinated macaques after SHIV challenge. The significance of difference was determined by log-rank test.

(Fig 2E). Although there was no obvious difference in the CD4+/CD8+ T cell ratio between vaccinated and controls (Fig 2F), all vaccinated macaques had better clinical outcomes as demonstrated by gradual weight gain (Fig 2G, left), which was in great contrast to control macaques. During the course of infection, 67% (6/9) control animals showed marked weight loss and/or persistent diarrhea and were euthanised (Fig 2G, right). Importantly, the pRhPD1-p27 vaccination conferred a significant survival advantage over controls (Fig 2H). These results demonstrated that the PD1-based vaccination strategy achieves significant suppression of $SHIV_{SF162P3CN}$ infection, starting as early as the acute phase and persisting into the setpoint phase.

## The pRhPD1-p27 vaccination induced poly-functional effector-memory T cells in rhesus macaques

To determine the adaptive immunity induced by pRhPD1-p27 for protection, Gag-specific humoral and cellular responses were measured in control macaques. Anti-Gag IgG antibodies were elicited by the homologous pRhPD1-p27 vaccination (S2 Fig). Similarly, anti-p27 T cell response was induced and boosted by the consecutive pRhPD1-p27 vaccinations as measured by ELISpot (Fig 3A). At 2–3 weeks post-last immunisation, the number of spot-forming units (SFU) in group A reached the level induced by the heterologous $MVTT_{SIVgpe}$ prime and $AD5_{SIVgpe}$ boost vaccine regimen as we previously described [12] (Fig 3B). Furthermore, the SFU in group B was higher than that in group A or in the heterologous regimen. To study the cellular responses in greater detail, *ex vivo* intracellular cytokine staining (ICS) analysis was conducted. An average of 1.2% of CD8+ and 0.6% of CD4+ T cells were specific to the Gag-p27 capsid antigen (Fig 3C–3E). The majority of the responding T cells were polyfunctional, producing both IFN-γ and tumour necrosis factor α (TNF-α), while some also produced interleukin-2 (IL-2). During the memory phase, evaluated at 22 weeks post-last immunisation, the p27-specific 0.36% of CD8+ and 0.14% CD4+ T cells remained detectable and remained polyfunctional (Fig 3F). A higher frequency of p27-specific CD8+ T cells that produced IFN-γ and TNF-α was elicited in group B as compared to group A. We next determined the memory phenotypes of the pRhPD1-p27-induced memory T cells using the markers CD28, CD95, and CCR7 [31,32]. Compared to the overall CD8+ or CD4+ compartments, most IFN-γ-producing cells were CD95hiCCR7loCD28hi/lo, indicating that the p27-specific T cells were enriched for the effector-memory 1 and effector-memory 2 T cell subsets (Fig 3G–3H).

## pRhPD1-p27-induced T cells in vaccinated macaques targeted various regions of the capsid antigen

As the breadth of Gag-specific T cell responses is associated with better HIV-1 control [26,33], we examined the T cell epitope profiles of the pRhPD1-p27-vaccinated macaques. ELISpot

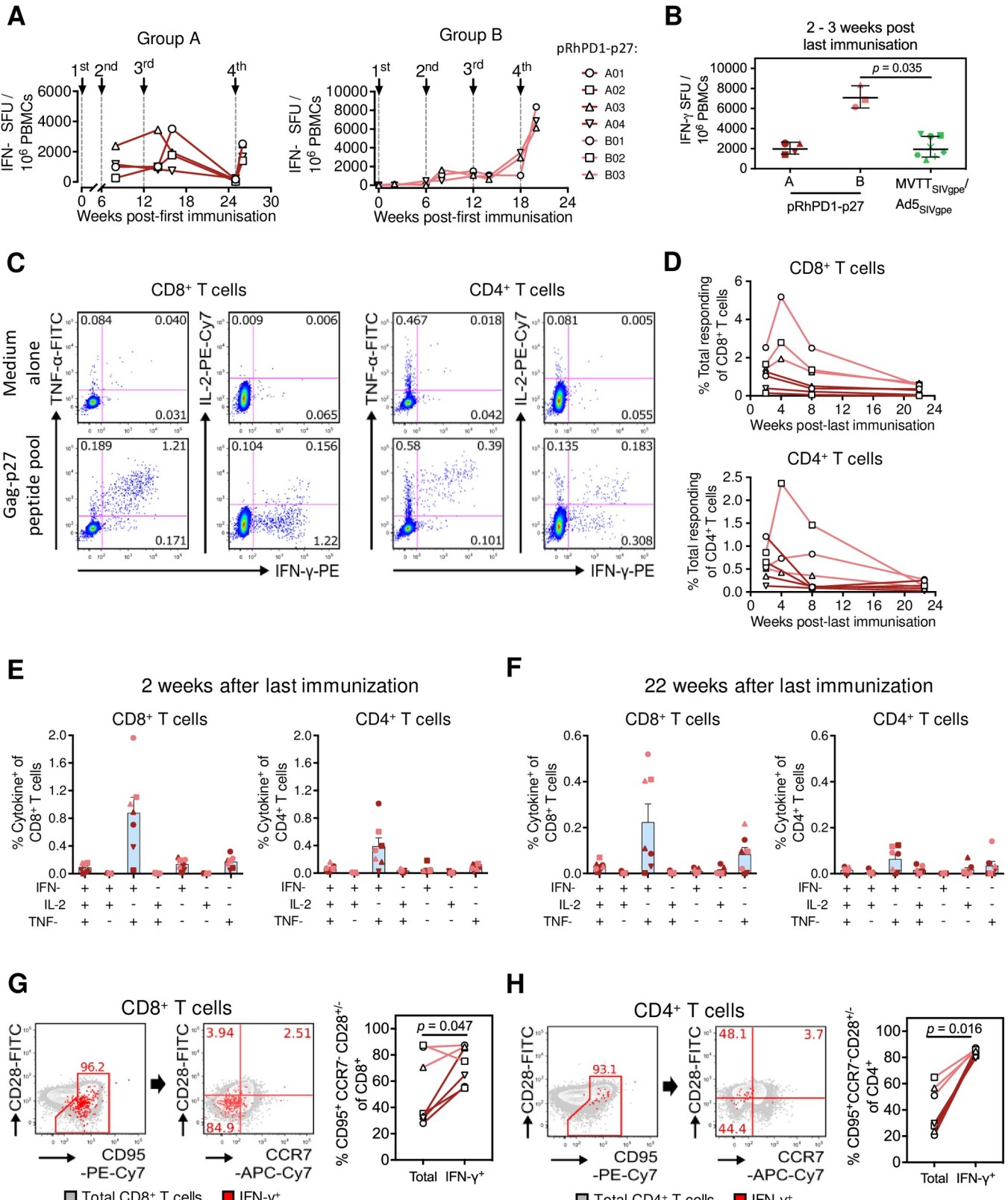

**Fig 3. T cell immunogenicity of the pRhPD1-p27 vaccine in rhesus macaques.** (**A**) Kinetics of p27-specific cellular immune responses in peripheral blood measured by IFN-γ ELISpot. (**B**) Comparison of ELISpot-detected p27-specific T cell responses induced in Groups A and B by the pRhPD1-p27 vaccine with

historical data of Gag-specific T cell responses induced by MVTT$_{SIVgpe}$/AD5$_{SIVgpe}$ vaccination in Chinese-origin rhesus macaques [12]. **(C)** Representative FACS plots demonstrating the production of IFN-γ, TNF-α and IL-2 by CD8+ (left) and CD4+ T cells (right) at 2 weeks post last immunization, as detected by ICS assays after *ex vivo* stimulation with the overlapping SIV Gag-p27 peptide pool. **(D)** Kinetics of p27-specific CD8+ (top) and CD4+ T cells (bottom) in PBMCs after last immunization, as detected by ICS assays. **(E)** Poly-functionality analysis of p27-specific CD8+ (left) and CD4+ T cell responses (right) at 2 weeks after last immunisation. **(F)** Poly-functionality analysis of CD8+ (left) and CD4+ T cell responses (right) during memory phase at 22 weeks after last immunisation. For (E) and (F), means ± SEM are shown. **(G)** Phenotypes of p27-specific CD8+ and **(H)** CD4+ T cells induced by pRhPD1-p27 vaccination in macaques. PBMC isolated at 8 weeks post last immunisation were stimulated with the overlapping Gag-p27 peptide pool, followed by surface and ICS analysis. Representative FACS plots (left) show the expression of CD95, CD28, and CCR7 on the overall T cell populations (grey) and IFN-γ-producing T cells (red). Frequencies of T$_{EM}$ (CD95+ CD28$^{+or—}$CCR7$^-$) were compared between the overall and IFN-γ-producing T cells (right). Statistical significance was determined by the Wilcoxon matched-pairs signed rank test.

assays were first performed to determine the pRhPD1-p27-induced T cell responsiveness towards individual overlapping 15mer peptides spanning the entire SIV Gag-p27 capsid antigen (Fig 4A). ICS was then conducted to confirm the reactive epitopes individually (Fig 4B). The number of mapped CD4+ and CD8+ T cell epitopes for each pRhPD1-p27-vaccinated macaque ranged from 5 to 7 (Fig 4C), in line with those elicited by adenoviral/poxviral vectors expressing mosaic HIV antigens detected in the Gag-p24 region [18]. Furthermore, the epitopes mapped did not confine into a particular region but were found along the length of the Gag-p27 antigen. The variation was expected, given that outbred Chinese-origin macaques were studied to model the human genetic diversity.

We also assessed major histocompatibility complex (MHC) restriction of the T cell epitopes from the vaccinated macaques of group B using blocking reagents in ICS (S3 Fig). The MHC class-I blocking antibody used at a concentration of 1.5mg/ml did not produce obvious changes in CD8+ T cell responses against the mapped epitopes. The MHC class-II blocking antibody inhibited T cell responses against the CD4+ T cell epitopes from the two macaques tested, but did not affect responses towards the CD8+ T cell epitopes. Pre-incubation with the MHC class-E-blocking VL9 peptide [34] also did not reduce epitope-reactive CD8+ T cell responses. Therefore, it was unlikely that the PD1-based vaccine elicits non-canonical MHC-restricted CD8+ T cell responses.

## Strong anamnestic capsid-specific T cell responses were recalled in pRhPD1-p27-vaccinated macaques after high-dose intravenous SHIV$_{SF162P3CN}$ challenge

Immunological recall responses were investigated after SHIV$_{SF162P3CN}$ challenge to determine how pRhPD1-p27-induced memory immunity performed against the pathogenic virus. Control macaques failed to mount anti-viral T cell responses above background levels, even up to 17 weeks post-challenge (Fig 4D–4F). In contrast, SHIV$_{SF162P3CN}$ infection resulted in strong anamnestic CD8+ and CD4+ T cell responses against Gag-p27 in pRhPD1-p27-immunised macaques (Fig 4D–4F). There was a trend for higher p27-specific CD8+ T cell responses in group B than in group A at both 4 and 17 wpi. Importantly, p27-specific CD8+ T cells in group B were robustly expanded, reaching a frequency up to 13.3% of peripheral CD8+ T cells at 4 wpi. *De novo* Nef-specific T cell responses were generated in 6 of the 7 vaccinated macaques at 17 wpi but could only be detected in one of the three surviving control macaques (Fig 4E and 4F). To identify potential immune correlates of viremia control, we measured neutralizing antibodies, viral loads and T cell responses and protective MHC class-I alleles in the vaccinated macaques. Both the vaccinated and control macaques showed barely autologous neutralising antibody responses after SHIV$_{SF162P3CN}$ challenge (S4 Fig). A trend for better peak viremia control was found in macaques with higher vaccine-induced p27-specific CD8+ T cell responses (S5 Fig, left). A similar trend was detected between peak viremia control and anamnestic p27-specific CD8+ T cell responses after SHIV$_{SF162P3CN}$ challenge (S5 Fig, right). These trends, however, did not reach

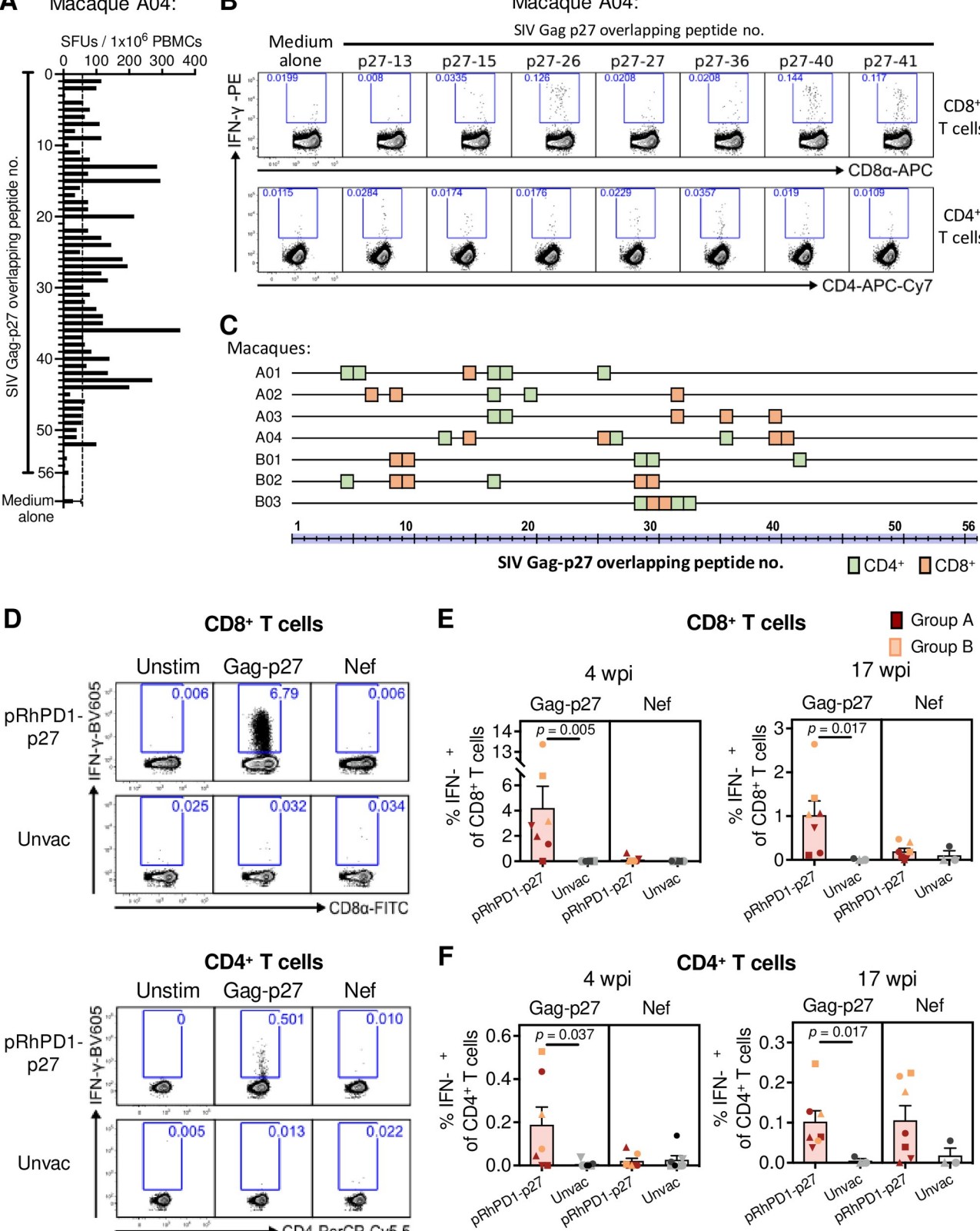

**Fig 4. Specificities of the T cell responses induced by the pRhPD1-p27 vaccine.** Representative results of T cell epitope mapping based on the IFN-γ ELISpot and ICS assays on macaque A04 from Group A. **(A)** PBMC isolated after third/fourth vaccination were stimulated with individual overlapping

15mer peptides spanning the Gag-p27 antigen in IFN-γ ELISpot assays. The dotted line represents the cut-off value. **(B)** Peptides that successfully induced responses above cut-off values (2x medium alone values) in IFN-γ ELISpot assays were then tested for their ability to induce IFN-γ production in T cells by ICS assays. **(C)** Epitope mapping analysis of the pRhPD1-p27-vaccinated macaques. CD8$^+$ and CD4$^+$ T cell epitopes are represented by orange and green boxes respectively. **(D)** Representative ICS FACS plots showing IFN-γ expression in CD8$^+$ (upper) and CD4$^+$ T cells (lower) from pRhPD1-p27-vaccinated or unvaccinated macaques 4 weeks after SHIV$_{SF162P3CN}$ challenge, after *ex vivo* stimulation with SIV p27 or Nef peptide pools. **(E)** Frequencies of CD8$^+$ and **(F)** CD4$^+$ T cells specific to Gag-p27 and Nef antigens at 4 or 17 weeks after SHIV$_{SF162P3CN}$ challenge, as determined by ICS analysis. Means ± SEM are shown. Significance of difference was determined by the two-tailed Wilcoxon rank sum test.

statistical significance. MHC class-I genotypes was then determined by deep sequencing. None of the vaccinated macaques carried the protective Mamu-A1*001 [35] or Mamu-B*008 alleles [36], but two in group A (A02 and A04) expressed Mamu-B*017 (S1 Table).

Next, we determined whether the T cell epitopes recognised during SHIV$_{SF162P3CN}$ infection in vaccinated macaques were similar to those induced by the PD1-based vaccine. This is an important parameter that would demonstrate the direct involvement of vaccine-induced T cell responses against viral challenge, yet only few studies determined the epitopes involved in anamnestic T cell responses primed by AIDS virus vaccine candidates during viral challenge in out-bred macaques. Since limited samples were available, we managed to test one previously-mapped CD4$^+$ and one CD8$^+$ T cell epitopes. At 17 weeks post SHIV$_{SF162P3CN}$ infection, 2 and 3 out of the 4 macaques in group A had CD8$^+$ and CD4$^+$ T cells that recognised epitopes identified during the vaccination phase, respectively (Fig 5A and 5B). We also conducted in-depth epitope mapping analysis using the peptide library spanning the p27 capsid antigen with samples from group B after viral challenge. The results showed that T cells responding to the SHIV$_{SF162P3CN}$ challenge from the vaccinated macaques were reactive towards similar sets of epitopes induced by the pRhPD1-p27 vaccination (Fig 5C). Taken together, the immunological analyses suggested that pRhPD1-p27-induced broad memory T cell responses were recalled upon SHIV$_{SF162P3CN}$ infection for viremia control.

## CD8$^+$ T cells mediated viremia control as demonstrated by viral rebound after *in vivo* CD8$^+$ T cell depletion

To assess the role of vaccine-induced CD8$^+$ T cells on viral suppression, CD8$^+$ T cells were depleted from the vaccinated macaques in group A, which displayed undetectable viremia at 17 wpi, using the anti-CD8β depleting antibody CD8b255R1. A single intravenous injection of CD8b255R1 led to a moderate CD8$^+$ T cell reduction in peripheral blood (S6A Fig) but resulted in rapid pVL rebound that peaked at 5–7 days after antibody treatment in three of the four tested macaques (Fig 5D). Viremia returned to undetectable levels within 3 weeks after antibody treatment. These findings highlight the crucial role of CD8$^+$ T cells in mediating the sustained viral control achieved after the RhPD1-p27 vaccination. A trend of a correlation between the level of CD8$^+$ T cell depletion and the magnitude of viral rebound was observed (S6B Fig). We continued to monitor the pVL in group A (Fig 5D). Three of the four vaccinated macaques maintained undetectable pVL for more than 90 weeks. One macaque, A01, showed a transient viral blip at 55–65 wpi, which subsequently declined to undetectable level. Plasma viral RNA obtained from this animal during the viral blip was sequenced in the Gag-p27 region. No changes in the amino acid sequence were detected within this region from all the 9 amplified samples (S6C Fig), suggesting that immune escape mutations had not been selected.

## Boosting immunisation of vaccinated macaques with sustained viremia control did not activate productive viral rebound

At 2 years post-SHIV$_{SF162P3CN}$ challenge, macaques from group A were re-immunised with the pRhPD1-p27 vaccine to test for the immunological and virological outcomes of the

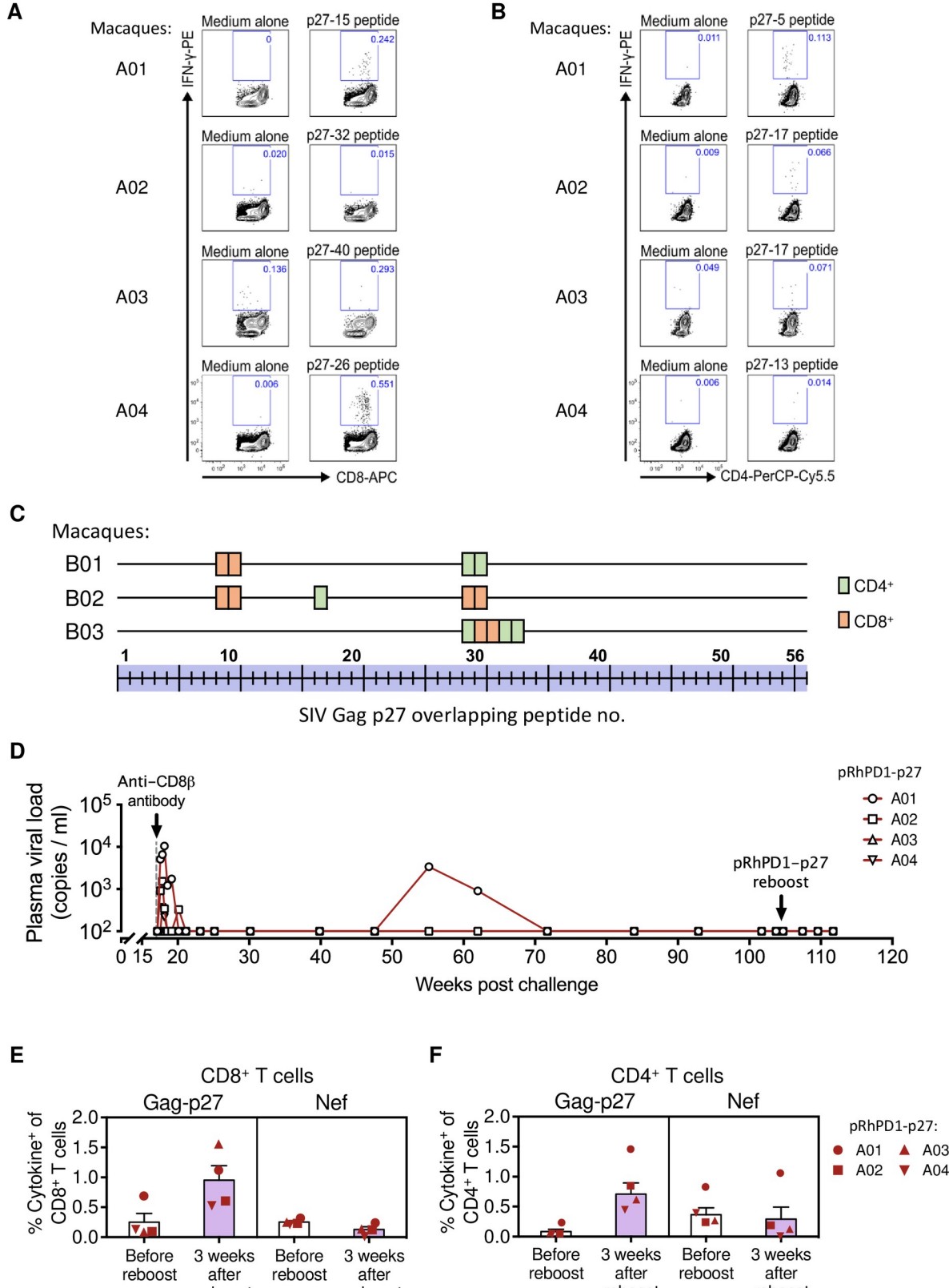

**Fig 5. Reactivity of T cell epitopes induced by the pRhPD1-p27 vaccine during SHIV$_{SF162P3CN}$ challenge. (A)** T cell responses against the PD1-based vaccine-induced CD8+ and **(B)** CD4+ T cell epitopes at 17 weeks after SHIV$_{SF162P3CN}$ challenge. **(C)** T cell epitope profiles

on Gag-p27 of vaccinated macaques in Group B after SHIV$_{SF162P3CN}$ challenge. **(D)** Long-term pVL monitoring in the vaccinated/ challenged macaques from Group A. Anti-CD8β antibody (clone CD8b255R1) was infused into these vaccinated macaques intravenously at 17 weeks post-challenge, after confirming pVL had reduced to undetectable levels. At 103 weeks post-SHIV$_{SF162P3CN}$ challenge, these macaques also received a boost immunisation of the pRhPD1-p27 vaccine via i.m./EP. **(E)** Frequencies of CD8+ and **(F)** CD4+ T cells specific to Gag-p27 and Nef antigens 3 weeks after pRhPD1-p27 re-immunization. Means ± SEM are shown.

PD1-based vaccine as a boost vaccine. Re-immunisation with pRhPD1-p27 did not result in measurable pVL based on multiple tests over time following re-immunisation (Fig 5D) but enhanced primarily p27-specifc CD8+ and CD4+ T cell responses while infection-induced Nef-specific T cell responses remained unaffected (Fig 5E and 5F). These results demonstrated that the PD1-based DNA vaccination can boost anti-viral memory T cell responses after 90 weeks without activating viral rebound in infected macaques, which would be useful for functional cure.

## Vaccine-induced expansions of three effector memory CD8+T cell subsets

To understand how the pRhPD1-p27 vaccination boosted CD8+ T cell memory responses significantly by the last vaccination, we performed single-cell RNA sequencing (scRNAseq) analysis to delineate CD8+ T cell profiles of macaques B01, B02, and B03 before last immunisation, 4 weeks post, and 12 weeks post last immunisation in group B. Clustering analysis using Uniform Manifold Approximation and Projection (UMAP) revealed 8 distinct clusters, and these clusters were annotated using SCSA according to cell markers published in CellMarker for identifying cell types (Fig 6A and 6B) [37,38]. We also confirmed the key markers of each cell type based on markers published in the literature. Different CD8+ T cell subsets, including 3 different effector memory CD8+ T subsets (Cluster 5, 7 and 8), were identified with high expressions of cytotoxicity related genes eg. *GNLY*, *TYROBP* or granzyme related genes (Fig 6A and 6B, S2 Table). When investigating the dynamic of different CD8+ T cell clusters, we revived that the effector memory CD8+ T cells Cluster 7 and 8 were significantly expanded in Macaque B01 and B03 four weeks after last immunization, while Cluster 5 were expanded in Macaque B02 and B03 (Fig 6C). Trajectory inference that measures dynamics of cell progression based on transcriptome similarities of cells also indicated that gene expressions of CD8+ T cells changed in individual animals after receiving vaccination with cells differentiated into different effector CD8+ T cell subsets (Fig 6C). We next evaluated differentially expressed genes (DEGs) in CD8+ T cells that possessed cytotoxic characteristics i.e. Cluster 5 to 8 after vaccination. We found very few DEGs for Cluster 7 although 2 of them were related to cytokine or immune receptors pathways when performing gene ontology (GO) analysis (S7 Fig). Instead, we found DEGs related to biological process highly expressed and enriched from Cluster 5, 6 and 8, and many of them were related to immune pathways. The GO analysis showed that the top 3 pathways that DEGs related to were 1) immune response, 2) regulation of immune system process and 3) regulation of immune response. (Fig 6D and S3 Table). In particular, BST2 were expressed in Clusters 5, 6 and 8 with Cluster 6 also highly expressed different MHC *Mamu* class I alleles (S3 Table). We also found that these cytotoxic CD8+ T cell clusters expressed DEGs related to molecular functions and cellular component (S7 Fig). Taken together, PD1-based vaccination expanded three effector memory CD8+ T cell subsets that expressed gene signatures important in mediating antigen recognition and cytotoxic functions during viral challenge.

## Discussion

In this study, we showed that the PD1-based DNA vaccine pRhPD1-p27 was immunogenic in rhesus macaques after intramuscular electroporation delivery. The homologous vaccination

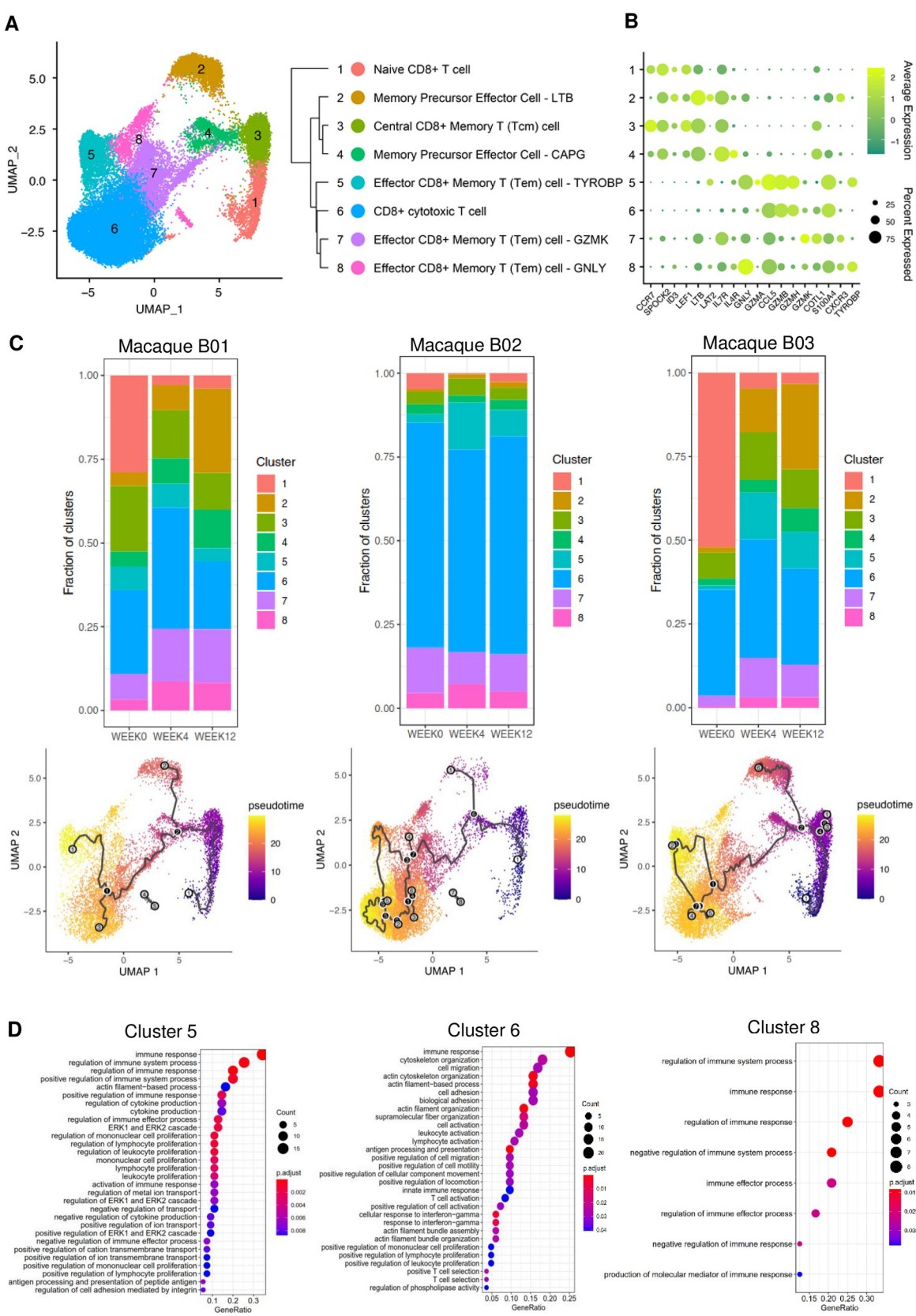

**Fig 6. Single-cell RNA sequencing analysis of CD8+ T cells induced by pRhPD1-p27 immunisation.** CD8+ T cell from macaques B01, B02, and B03 in Study B were purified at before, 4 weeks post, and 12 weeks post last immunisation followed by scRNAseq using the 10X scRNAseq platform. (**A**) UMAP clustering analysis revealed 8 major CD8+ T cell clusters. (**B**) Expression of different marker genes of the indicated CD8+ T cell clusters. (**C**) Fractions of different CD8+ T cell clusters at different time points after vaccination (top) and trajectory inference (below) showing the dynamic of cell progression after receiving vaccination of individual macaques. (**D**) GO analysis of differentially expressed genes of different pathways related to biological process. GeneRatio represents the ratio of the number of genes related to the GO term to the total number of significant genes.

regimen not only induced primarily polyfunctional effector memory CD8+ T cell responses, but also resulted in sustained viremia control and significantly prolonged survival after a high-dose intravenous SHIV$_{SF162P3CN}$ challenge in two separate experiments consisted of seven vaccinated macaques (100%, 7/7). This was in great contrast to most control animals that developed simian AIDS (67%, 6/9). A decrease in peak viremia was also observed in the experiment with the shortened vaccination schedule. Mechanistically, vaccine-induced effector-memory CD8+ T cells were recalled effectively upon viral challenge for protection as determined by T cell epitope mapping and CD8+ T cell depletion experiments. Furthermore, vaccinated macaques with sustained viremia suppression for over two years responded to a boost vaccination without leading to viral reactivation or rebound. These promising results may warrant the clinical development of the PD1-based DNA vaccine as a potential immunotherapy for functional cure, given the straightforward and cost-effective process for producing clinical grade DNA vaccines and feasible DNA vaccine delivery via electroporation in clinical trials [39,40].

PD1-based antigen DC-targeting is an innovation for enhancing the induction of effector-memory CD8+ T cells in NHP. Other DC-targeting strategies have previously been explored in HIV/SIV vaccines in NHP. One of the well-studied approaches involves fusion of antigens to an anti-DEC205 antibody. The HIV-1 capsid protein Gag-p24 fused to anti-DEC205 antibody elicited primarily CD4+ T cell responses, but not CD8+ T cell responses, in NHP [20]. Similarly, vaccination of rhesus macaques with anti-DEC205-tagged antigen of malaria circumsporozoite also elicited CD4+ T cell-biased responses even with the use of poly(I:C) as an adjuvant [41]. In a more relevant study of using the same test antigen, a DNA vaccine encoding a recombinant anti-DEC205-fused SIV Gag-p27 capsid antigen failed to induce an effective T cell response in NHP [42]. Moreover, a combination vaccination regimen using the DEC205-targeting DNA priming and SIV-based virus-like particle boosting did not confer significant protection against intrarectal viral challenge [43]. These data suggest that DEC205-based antigen DC-targeting may not be a suitable strategy for priming CD8+ T cell immunity. Other DC-targeting antibodies recognising LOX-1, CD40, and DCIR have been included in priming or boosting vaccines tested in NHP with similar results [21,22]. These antigen DC-targeting strategies yielded relatively low CD4+ T cell-centric cellular responses and their protective efficacy has not been determined. The PD1-based vaccine, therefore, may be unique in eliciting antigen-specific effector-memory CD8+ T cells in NHP, which is consistent to our previous findings in mice [24].

Sustained viremia control underlying PD1-based vaccination may involve polyfunctional effector-memory CD8+ T cells for protection. Currently, the most promising AIDS vaccine candidates in conferring sustained viremia control against pathogenic SIV/SHIV infections involved viral-vectored T cell vaccines such as RhCMV-based vaccine [8,9,11], heterologous Ad5/26 adenovirus-based vaccine [10,18,19] and heterologous vaccinia prime and adenovirus boost vaccine [12]. In these studies, protection was encouragingly observed in subgroups of the vaccinated macaques with complex mechanisms involved. For example, the RhCMV-based vaccine achieved prolonged viremia control in over 50% of vaccinated Indian macaques by inducing exceptionally broad and unconventional MHC E-restricted T cell responses [34,44]. The heterologous rAd26 prime rAd5 boost vaccine regimen conferred improved

protection by eliciting higher frequency of polyfunctional T cell immunity in vaccinated Indian macaques [14]. Our PD1-based vaccine induced CD8[+] T cell response against a similar breadth of T cell epitopes as to the heterologous rAd26 prime rAd5 boost vaccine regimen [16,18], but was narrower than the RhCMV-based vaccine [34,44]. Nevertheless, the PD1-based vaccine achieved prolonged viral suppression in all vaccinated macaques. Importantly, our epitope mapping analysis reveals that the PD1-based vaccine successfully induces functional T cell responses that are responsive towards SHIV$_{SF162P3CN}$ infection in the out-bred macaques examined. This result demonstrates the direct involvement of vaccine-elicited T cells during viral challenge. Compared to the heterologous vaccinia prime and adenovirus boost vaccination strategy that we examined previously [12], the homologous PD1-based DNA vaccination has the potential to prime a stronger anti-viral T cell response, providing an optimized vaccination schedule is followed.

Other vaccine-induced protective T cell responses focused on well-defined SIV epitopes in MHC-determined macaques [45,46]. For example, Mamu-A*01, Mamu-B*08 and Mamu-B*017 may model some elite controllers of HIV-1 infection in Indian macaques with better chronic viremia control [36,47]. However, we saw neither exceptionally broad and unconventional MHC class II/E-restricted T cell responses, nor a particular protective MHC class I allele. Only two of our seven vaccinated macaques had Mamu-B*017. The protective role of this MHC allele remains questionable because our pRhPD1-p27 antigen was not targeted by Mamu-B*017-restricted CD8[+] T cells [48,49].

The scRNAseq results showed that three effector memory CD8[+] T cell subsets were expanded by homologous pRhPD1-p27 vaccinations with high expressions of cytotoxicity related genes including *GNLY*, *TYROBP* (i.e DAP12), *HCST* (i.e DAP10) and granzyme related genes, indicated that these cells might display a highly cytotoxicity phenotype [50–52]. In fact, trajectory inference of CD8[+] T cells indicated that pRhPD1-p27 vaccination drove the differentiation and expansion of naïve or precursor cells (Cluster 1, 2 and 4) into different effector CD8[+] T cell subsets (Cluster 5 to 8). It was likely that these effector cells were important in mounting specific immune response against the p27 antigen upon immunization, thereby launching effective viral clearance upon infection. Enhanced expression of GNLY in CD8[+] T cells has been reported in humans vaccinated with a live attenuated shingles vaccine and in antigen-specific T cells in humans vaccinated with yellow fever vaccination [53,54] It will be of interest to further evaluate clonal expansion of these subsets to more specifically characterize key CD8[+] T effector cell subsets that potently recognize the p27 antigen. This will facilitate our understanding on how the PD1-based DNA vaccine mediates memory T cell responses against the p27 antigen after these clonally expanded effector T cells return to resting memory state and again encounter the virus during challenge.

Overall, the induction of high frequent polyfunctional effector-memory CD8[+] T cells is likely a commonly shared mechanism in both Indian and Chinese macaques contributing to prolonged viremia control in many studies including the RhCMV-based vaccine [8,12,14,45]. It is encouraging that repeated PD1-based vaccinations resulted in three different effector memory CD8[+] T cell populations with high cytotoxicity-related gene expression and that effector-memory CD8[+] T cells induced by the PD1-based vaccination were recalled effectively after pathogenic viral challenge in our out-bred macaque cohort consisted of diverse MHC alleles. Moreover, in controller macaques PD1-based vaccination recalled mainly capsid-specific CD8[+] T cell responses that had previously received the same vaccine without causing viral rebound. Future studies should evaluate the therapeutic potential of the PD1-based vaccine following an annual vaccination regimen in infected macaques. To provide direct evidence for functional cure of HIV-1 infection in humans, however, the immunogenicity and efficacy of PD1-based vaccine strategy should be carefully evaluated in human trials.

## Materials and methods

### Ethics statement

All mouse experiments were conducted at the University of Hong Kong and were approved by its Committee on the Use of Live Animals in Teaching and Research. Female C57BL/6 mice at the age of 6–10 weeks were used. Our macaque studies were approved by the Institutional Animal Care and Use Committee of the Department of Veterinary Medicine at the Foshan University. For both mice and macaques, animal welfare and condition of all animals were closely monitored according to approved protocol. Animals were euthanized when the experimental endpoint was reached. Outbred Chinese-origin, female rhesus macaques that were purpose bred were housed at the Foshan University Animal Research Centre and were free of simian immunodeficiency virus infection before the start of the vaccination regimen. Corresponding identity numbers used in the animal facility for individual macaques were shown in S4 Table. MHC-I genotypes of the vaccinated macaques were determined by MiSeq (Illumina) based on the method by Wiseman et al [55] with modifications. In brief, amplicons covering the MHC-I exon two region were generated from cDNA that was reverse transcribed from RNA isolated from peripheral blood mononuclear cells (PBMC) using primers 5'-GAAGTGATTACGGTG TGCGCCTCGCTCTGGTTGTAGTAG-3' and 5'-GAGTTGGATGCTGGATGGGGGCTAC GTGGACGACAC-3'. The amplicons were then sequenced using a MiSeq platform (250bp paired-end; performed by Genewiz). The sequence reads were then compared against all known MHC-I alleles from the Immuno Polymorphism Database—MHC Database [56] for MHC-I mapping.

### Vaccine constructs

Codon-optimized DNA constructs of the sPD1-fused Gag-p55 antigen and the sPD1-fused Gag-p27 capsid antigen were synthesized and cloned into the pVAX plasmid vector (Invitrogen) to generate pGag, pRhPD1-Gag and pRhPD1-p27 DNA vaccines. Expression of the immunogens were confirmed by western blotting using the anti-HIV-1 Gag-p24 antibody clone 13-H12-5C (NIH AIDS Reagents Program) that is cross-reactive to the SIV-Gag p27 antigen.

### *In vitro* binding of sPD1-fused immunogens to PD1 ligands

293T cells were transfected with the pGag, pRhPD1-Gag or pRhPD1-p27 DNA vaccines using polyethylenimine. Three days later, supernatant was collected and were used as the sources of recombinant immunogens for testing their binding ability to PD1 ligands expressed on transiently-transfected 293T cells, as described previously [24].

### Immunization

Mice were immunized with 100 μg DNA vaccines indicated or PBS via im/EP three times at 3-week intervals. Macaques were either received 2 mg pRhPD1-p27 vaccine via im/EP four times at 6- to 13-week intervals or left unvaccinated, as indicated in (Fig 1A). The im/EP was done using the TERESA-EPT Gene Delivery Device ($2^{nd}$ generation) (Shanghai Teresa Healthcare Sci-Tech Co., Ltd.,).

### MVTT$_{SIVgpe}$ challenge in mice

Vaccinated mice were treated with $4 \times 10^7$ plaque forming units of vaccinia virus MVTT$_{SIVgpe}$ [12] via intranasal route two weeks after last immunisation. Challenged mice were euthanised

two days later and viral loads were determined in the lungs using standard plaque assays on Vero cells.

## SHIV$_{SF162P3CN}$ challenge in macaques

The SHIV$_{SF162P3CN}$ challenge stock was generated by passage and expansion of the parental SHIV$_{SF162P3}$ strain in IL-2-treated Chinese-origin rhesus macaque PBMC that were depleted of CD8+ T cells by magnetic bead separation (Miltenyi Biotec). TCID$_{50}$ of the SHIV$_{SF162P3CN}$ stock was determined with TZM-bl cells using luciferase reporter assays [57]. Chinese-origin rhesus macaques were challenged with 5000 TCID$_{50}$ of SHIV$_{SF162P3CN}$ intravenously via saphenous vein. CD4+ and CD8+ T cell count were determined by flow cytometry or using Trucount tubes (BD Biosciences).

## *In vivo* CD8+ T cell depletion

Macaques were treated with a single intravenous injection of 50mg/kg body weight of anti-CD8β depleting antibody CD8b255R1 based on the standard protocol provided by the vendor (NHP Reagent Resource).

## Viral quantitation assays

Viral RNA isolated from plasma using a QIAamp Viral RNA Minikit (Qiagen) was reversed transcribed into cDNA using a PrimeScript RT Reagent kit (Takara) with random hexamers. Plasma viral RNA copy number was then determined with a SYBR green-based real-time quantitative PCR (Takara) using SIV-Gag-specific primers 5'-GTAGTATGGGCAGCAAAT GAAT-3' and 5'-CACCAGATGACGCAGACAGTAT-3' [58]. To determine cell-associated DNA viral load, DNA was firstly isolated from PBMC using a QIAamp DNA Mini Kit (Qiagen). Digital PCR was then performed on the QuantStudio 3D Digital PCR system (Thermo Fisher) to determine proviral loads, with primers 5'-GTCTGCGTCATYTGGTGCATTC-3' and 5'-CACTAGYTGTCTCTGCACTATRTGTTTTG-3', and probe 5'-FAM-CTTCRTCAG TYTGTTTCACTTTCTCTTCTGCG-BHQ1-3' [59]. The results were normalised based on albumin gene copy numbers determined by real-time quantitative PCR [60].

## Humoral immune assays

Gag-specific antibody responses in plasma were assessed by ELISA [61] against the SIV Gag-p55 antigen produced from transiently-transfected 293T cells. Neutralizing antibody activity against SHIV$_{SF162P3CN}$ was measured in heat-inactivated plasma from the infected macaques with TZM-bl cells using luciferase reporter assays [57].

## Cellular immune assays

T cell immune responses were examined by tetramer staining, IFN-γ ELISpot or intracellular cytokine staining assays. In tetramer staining, H-2D$^b$/AL11 tetramer (AL11 amino acid sequence: AVKNWMTQTLL; MBL International) was used to determine the frequency of CD8+ T cells specific for the immunodominant AL11 epitope from the SIV Gag-p27 antigen in vaccinated C57BL6 mice. In ELISpot assays, 1-2x10$^5$ splenocytes from mice or PBMC from rhesus macaques were stimulated with 1μg/ml of 15-mer overlapping peptide pools spanning the SIVmac239 Gag-p55 or Gag-p27 antigens (NIH AIDS Reagents Program), and the T cell responses were determined using mouse and macaque IFN-γ T cell ELISPOT kit (U-Cytech and Mabtech ) as previously described [12,61]. In ICS assays, up to 1x10$^6$ macaque PBMC were stimulated with 1 μg/ml of 15-mer overlapping Gag-p27 or Nef peptide pools in the

presence of 0.5 μg/ml of anti-CD28 and anti-CD49d antibodies (Biolegend). Backgrounds were determined in co-stimulated cells without the presence of antigens. After 2-hour incubation at 37°C with 5% $CO_2$, Brefeldin A (7.5 μg/ml; Sigma-Aldrich) was added, and the cells were further incubation overnight. Cells were then surface stained, fixed and permeabilised using Cytofix/Cytoperm kit (BD Biosciences), and stained for intracellular cytokines. The following antibodies were used: anti-CD3 (SP34-2; Horizon V450), anti-CD4 (OKT4; PerCP-Cy5.5), anti-CD8α (RPA-T8; APC, APC-Cy7, FITC), anti-CD28 (CD28.2; FITC, PE), anti-CD95 (DX2; PE-Cy7) anti-CCR7 (G043H7; APC-Cy7), anti-IFN-γ (B27; BV605, PE), anti-TNFα (Mab11; FITC), anti-IL-2 (MQ1-17H12; PE-Cy7). The antibodies were from BD Biosciences or Biolegend. Zombie Aqua fixable viability stain was used during the surface staining step to discriminate against dead cells. Data were acquired with a BD FACSAria III instrument, and analysed using FlowJo v9.9 software. Mouse ICS assays were performed as previously described [61]. In the epitope mapping experiments, responses towards individual 15-mers peptides (1 μg/ml) were determined using PBMC isolated after the third/fourth round of vaccination or at least 4 weeks after SHIV challenge using ELISpot, followed by ICS assays. Minimal T cell epitope numbers were then determined. If two adjacent peptides produced positive results for the same T cell subtype, it would be counted as one single minimal T cell epitope. In blocking experiments, PBMC were firstly pre-incubated with 1.5 mg/ml anti-MHC-I (W6/32; BioXcell), anti-MHC-II (L243; BioXcell), isotype control (BioXcell), or 100 μM VL9 peptide (VMAPRTLLL; GL Biochem) 37°C with 5% $CO_2$ for 2 hours prior to antigen stimulation.

## Viral sequence analysis

Viral Gag-p27 sequences were amplified from cDNA generated from viral plasma RNA using a nested PCR approach with Phusion DNA polymerase (Thermo Fisher). Primers for the first round of PCR were 5'-GGTTGCAGGTAAGTGCAAC-3' and 5'-CAATAGAATCATCAG CCCCTG-3'. Primers for the second round of PCR were 5'- CCTAGTGGTGGAAACAGG AAC-3' and 5'-GCAGAGTGTCCCTCTTTCC-3'. The final PCR product was purified and sub-cloned into pMD18-T plasmid via TA cloning following the manufacturer's instructions (Takara). The inserted sequences were determined by Sanger sequencing (BGI and Genewiz).

## Single-cell RNA sequencing (scRNAseq) and bioinformatics analysis

Live CD3+CD8+ T cells from 9 samples (3 macaques, 3 time points) were subjected to scRNA-seq analysis using the droplet-based 10X Genomics platform for library preparation followed by sequencing using NovaSeq 6000 (Illumina). The customized Macaca mulatta reference package was created by the mkref function in Cell Ranger software (version 3.1.0) using the Macaca mulatta reference genome from Ensembl (Mmul_8.0.1, release 97). Unique molecular identifier (UMI) counts were then generated by aligning raw FASTQ files to the customized reference tracscriptome using the count function of Cell Ranger. Cells and genes with the following threshold parameters were removed using the Seurat package (version 3.2.2) before proceeding to subsequent analysis [62]: Genes expressed in 10 cells or less, number of genes detected per cell < 250, UMI counts < 500, percentage of mitochondrial gene expression > 5%. Doublets were excluded by R package DoubletFinder (version 2.0.3) [63]. A total of 33498 cells were remained after quality control. Normalization was performed with the default method mentioned in Seurat. The integration of samples and correction of batch effect were performed using the fast version of the mutual nearest neighbors (fastMNN) method provided by batchelor package (version 1.2.4) [64]. Dimensional reduction was performed using the Uniform Manifold Approximation and Projection (UMAP) approach. FindClusters function from Seurat package was used to divide cells with 8 clusters obtained (resolution = 0.3).

Phylogenetic tree of the identified clusters was constructed by the BuildClusterTree function. Cell types of each classification were evaluated by SCSA [37].Wilcoxon Rank Sum test was performed using the FindAllMarkers function to identify cluster-specific genes (logFC>0.25). Cell trajectories were constructed with the Monocle 3 package (version 0.2.3). Gene ontology analysis was performed using clusterProfiler package [65]. The data are deposited in NCBI's Gene Expression Omnibus (GEO) and are accessible through GEO Series accession number GSE171783 (https://www.ncbi.nlm.nih.gov/geo/query/acc.cgi?acc=GSE171783).

## Statistical analysis

Statistical analyses of data were performed using Prism v7 (GraphPad). In mouse studies, significances of differences were determined by ANOVA, followed by Tukey multiple comparisons test. In macaque studies, two-tailed Wilcoxon rank sum test or Kruskal-Wallis test followed by the Dunn's multiple comparisons test were performed for the comparisons between two treatment groups or more than two treatment groups, respectively. Survival analysis was performed using log-rank test. Probability value ($p$) of smaller than 0.05 was considered statistically significant.

## Supporting information

**S1 Table. MHC class I genotyping of the vaccinated macaques.** MHC class-I genotypes was determined by deep sequencing to determine whether vaccinated macaques carried protective MHC class I allele.
(DOCX)

**S2 Table. Top 20 differentially expressed genes within each CD8[+]T cell clusters of Group B vaccinated macaques identified by single-cell transcriptome analysis.** Single-cell RNA sequencing was done for Group B macaques before last immunisation, 4 weeks post, and 12 weeks post last immunisation using PBMCs. Data analysis was done as described under Materials and Methods.
(DOCX)

**S3 Table. Differentially expressed genes under the top 3 immunology related pathways in Cluster 5, 6, and 8.** Differentially expressed genes were annotated by GO analysis as described under Materials and Methods.
(DOCX)

**S4 Table. Corresponding identification numbers of the tested animals from the animal facility.** Animal ID assigned in this study and its corresponding ID in the breeding animal facility or animal facility for animal experiments were shown.
(DOCX)

**S1 Fig. Schematic structure of three DNA vaccine candidates and their binding to murine PD-L1 (msPD-L1) and PD-L2 (msPD-L2). (A)** Three DNA vaccine candidates were constructed using pVAX as the expression vector. A pair of DNA vaccines, expressing SIVmac Gag-p55 antigen, either alone (pGag) or fused to rhesus soluble PD1 domain (pRhPD1-Gag), were generated. DNA vaccine, pRhPD1-p27, was also constructed to encode for a rhesus soluble PD1 domain fused to the Gag-p27 capsid antigen. A $(G_4S)_3$ linker sequence was placed in between soluble PD1 domain and the antigen in pRhPD1-Gag and pRhPD1-p27. All Gag antigens were placed under the CMV promoter and contained a human tissue plasminogen activator (tPA) secretory signal sequence to promote antigen secretion. All constructs were codon optimised for expression in mammalian cells. **(B)** Soluble proteins pRhPD1-Gag and pRhPD1-p27 expressed and released from transfected HEK293 cells were confirmed for

binding to msPD-L1 and msPD-L2, respectively.
(TIF)

**S2 Fig. Humoral immune response induced by pRhPD1-p27 in rhesus macaques after intramuscular electroporation.** Anti-Gag IgG antibody dilution titers were measured by ELISA in the plasma isolated from pRhPD1-p27-vaccinated rhesus macaques from Group A (A) and Group B (B) at time points indicated.
(TIF)

**S3 Fig. Determination of MHC-restriction of T cell responses by pRhPD1-p27 induced in group B using anti-MHC-I, anti-MHC-II antibodies or MHC-E blocking VL9 peptide.** PBMCs isolated at 10 weeks (**A** and **B**) or at 8 weeks (**C**) post-last immunization from the immunized macaques were firstly incubated with anti-MHC-I, anti-MHC-II antibodies, or MHC-E-blocking VL9 peptide for 2 hours at 37˚C with 5% $CO_2$. Individual Gag-p27 peptides corresponding to the mapped T cell epitopes were then added to the cells and incubated at 37˚C with 5% $CO_2$. 2 hours later, BFA was added two hours later. After overnight incubation, cells were washed and stained for surface markers, followed by fixation with 2% PFA and stained for TNF-α and IFN-γ in Perm/Wash buffer. T cell responses, as determined by TNF-$α^+$ and IFN-$γ^+$, were determined using FACS. Results were normalized against the non-blocking isotype or untreated controls.
(TIF)

**S4 Fig. Low neutralizing antibody titers against the autologous SHIV$_{SF162P3CN}$ in pRhPD1-p27-vaccinated (left) and unvaccinated (right) macaques after viral challenge.** Plasma from the infected macaques was samples in the indicated timepoint to test the neutralizing antibody activity against SHIV$_{SF162P3CN}$ with TZM-bl cells using luciferase reporter assay.
(TIF)

**S5 Fig. Correlation analysis of CD8+ T cell responses induced during the vaccination phase (left) and the challenge phase (right) vs viremia levels in the vaccinated macaques.** Vaccinated macaques from both groups A and B are shown in this analysis. *p* valves shown were calculated based on the Spearman rank-correlation test.
(TIF)

**S6 Fig. Outcomes of *in vivo* CD8+ T cell depletion using anti-CD8β depleting antibody CD8b255R1. (A)** Changes of peripheral CD8+ T cell frequency after intravenous injection of anti-CD8β depleting antibody CD8b255R1. **(B)** Correlation of peak viral load after CD8+ T cell depletion and the magnitude of CD8+ T cell depletion. *p* valves shown were calculated based on the Spearman rank-correlation test. **(C)** Sequence analysis did not show escape mutations in Gag-p27 encoded in the vaccine during viral load rebound.
(TIF)

**S7 Fig. Differential gene expression related to molecular function (left) and cellular component (right) of Cluster 5 to 8.** GeneRatio represents the ratio of the number of genes related to the GO term to the total number of significant genes. For Cluster 7, no DEGs were enriched in pathways related to cellular component.
(TIF)

**S1 Data. Raw data for main figures in manuscript.** Tables containing raw data of all main figures.
(XLSX)

**S2 Data. Raw data for supplementary figures in manuscript.** Tables containing raw data of all supplementary figures.
(XLSX)

**S3 Data. Group B vaccinated macaques scRNAseq metadata worksheet.** Descriptive information and protocols for experiments, samples and data processing pipeline of single-cell RNA transcriptome analysis.
(XLSX)

## Acknowledgments

We thank C. Cheng-Mayer for providing the parental SHIV$_{SF162P3}$ viral strain and critical discussions of the manuscript. We thank David Ho and Kwok-Yung Yuen for scientific advice. We thank Shanghai Teresa Healthcare Sci-Tech Co., Ltd., for providing im/EP technical support.

## Author Contributions

**Conceptualization:** Zhiwei Chen.

**Formal analysis:** Yik Chun Wong, Wan Liu, Lok Yan Yim, Xin Li, Hui Wang, Ming Yue, Mengyue Niu, Lin Cheng, Samantha M. Y. Chen, Haibo Wang, Xian Tang, Haoji Zhang, Youqiang Song, Lisa A. Chakrabarti, Zhiwei Chen.

**Funding acquisition:** Zhiwei Chen.

**Investigation:** Yik Chun Wong, Wan Liu, Lok Yan Yim, Xin Li, Hui Wang, Ming Yue, Mengyue Niu, Lin Cheng, Lijun Ling, Yanhua Du, Samantha M. Y. Chen, Haibo Wang, Xian Tang, Jiansong Tang, Haoji Zhang, Youqiang Song, Lisa A. Chakrabarti, Zhiwei Chen.

**Methodology:** Yik Chun Wong, Wan Liu, Zhiwei Chen.

**Resources:** Ka-Wai Cheung.

**Writing – original draft:** Yik Chun Wong, Lok Yan Yim, Zhiwei Chen.

**Writing – review & editing:** Lisa A. Chakrabarti, Zhiwei Chen.

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
