## [Decision Letter · Decision Letter 0]

16 Mar 2021

Dear Prof. Chen,

Thank you very much for submitting your manuscript "Sustained viremia suppression by SHIV-recalled effector-memory CD8+ T cells by PD1-based vaccine" for consideration at PLOS Pathogens. As with all papers reviewed by the journal, your manuscript was reviewed by members of the editorial board and by several independent reviewers. The reviewers appreciated the attention to an important topic. Based on the reviews, we are likely to accept this manuscript for publication, providing that you modify the manuscript according to the review recommendations.

Sincerely,

Daniel C. Douek

Associate Editor

PLOS Pathogens

Richard Koup

Section Editor

PLOS Pathogens

Kasturi Haldar

Editor-in-Chief

PLOS Pathogens

orcid.org/0000-0001-5065-158X

Michael Malim

Editor-in-Chief

PLOS Pathogens

orcid.org/0000-0002-7699-2064

Reviewer Comments (if any, and for reference):

Reviewer's Responses to Questions

**Part I - Summary**

Reviewer #1: This manuscript by Wong et al investigated a PD1-based vaccine by fusion of SIV capsid antigen to soluble PD1 in macaques model. They found induction of poly-functional CD8+ T cells by vaccination which were recalled upon viral challenge and contributed to viral control for over two years. This group in the past (Ref 24) has established a DC-targeting vaccine strategy by fusing HIV-1 antigen to a soluble PD-1, which capable of inducing robust CD8 T cell responses in mouse model. In this study, they took a step further and demonstrated the potency of such strategy in NHP. Overall I think this is a well-designed and performed study, the conclusion is well justified by results, which provided rational for an additional potential T cell based vaccine strategies.

Reviewer #2: This study is an extensive evaluation of the cellular immune responses to a gag-derived p27 vaccine targeted for Dendritic cells. The study is important and of broad interest to AIDS research community. Overall the study is a well-designed set of animal experiments, albeit small numbers, but results are sufficient to move this strategy forward toward eventual clinical trials. The cellular immune responses were impressive and are of high importance to the field of vaccine development for HIV/AIDS. Preliminary testing in mice using a vaccinia murine system allowed evaluation of vaccine efficacy before moving to macaques and should remain in the MS. pRhPD1-p27 showed better immune responses. For the rhesus experiment, passage of SHIVSF162p3 in Chinese origin rhesus PBMCs added confidence to using this stock for challenge.

The methods and approach to animal experiments were strong. The major components for a strong vaccine argument are the challenge virus, the challenge dose and the route of inoculation. In this study, all 3 components were a stringent test of the candidate vaccine in this macaque model. The 5000 TCID50 challenge dose by the IV route is among the strongest of challenges used in macaque models. Suppression of PVL was significant. The difference in clinical outcome for vaccine and control groups were significant, based on weight loss. Cellular responses to p27 were well documented in figs 3-5. Finally, CD8 depletion documented CD8 cell control.

Specific points

Results

1. Fig 2 – the numbers are small but the large differences between A and B vaccine groups are in favor of the short course vaccination and challenge. Viewing the stats in Fig. 2C, 2D and 2E strengthen conclusion of efficacy. Clinical evaluations, using weight loss, is strong component for clinical outcome. Were necropsies done? A table of the findings in supplementary data would add weight to the clinical conclusions drawn.

2. Line 151 The main points of the rest of the paper refer to CMI responses, and rightly so since p27 antibody is not known to play a role in control. But, what were the anti-gag IgG antibody responses of the other two groups. Was group B also positive? Group C should be negative Fig. S2 only shows group A.

3. In the interest of clarity, the macaque numbers given can cause issues with record keeping and data identification. Are these monkey numbers the house numbers from Foshan University Dept. of Veterinary Medicine? If not, can the animal house numbers be used?

4. Outbred Ch Rh lines 182 and 400 – Were outbred Ch Rh purpose bred or wild-caught? This point may seem trivial, but could be important for follow-up studies to reproduce results with larger numbers and move the program forward to clinical studies.

Discussion

1. I don’t see anything that is an overreach from the data.

**Part II – Major Issues: Key Experiments Required for Acceptance**

Reviewer #1: None

Reviewer #2: Changes will improve the clarity of the MS and data presentation, but are discretionary. Only point 4 is mandatory because it will help to protect the authors for verification of these preclinical studies. Wild caught and purpose bred animals are very different because of the environments and their immune systems can vary. Chinese researchers may use more wild-caught than purpose bred. In the USA, rhesus are almost all purpose bred animals. For reproducing these experiments, animals should be from the same background type or at least researchers should be cognizant of the differences. I am not implying that this comment invalidates anything, it does not, but reproducibility using animals with very different backgrounds can be a problem. Researchers in the field should be made aware of potential differences

**Part III – Minor Issues: Editorial and Data Presentation Modifications**

Reviewer #1: I have a few comments as the following:

1. Can author speculate if there are any quality or quality difference in T cell generated from other already established T cell based Vaccine Vaccine strategy in NHP?

2. Three effector memory CD8 identified in scRNAseq is interesting, but it is unclear if this is unique from T cell induced by natural infection or other T cell based Vaccination strategy.

Reviewer #2: 1. Overall well written, easy to follow. There are some syntax/grammatical issues here and there for example Line 402 – Welfares and conditions of all animals…should be, Animal welfare and condition of all animals…

2. line 113 Chinese Rhesus are a sub-species of rhesus macaque group, not a different species. Recommend for line 111 …first passaged the pathogenic SHIVSF162P3 strain isolated from an Indian-origin rhesus macaque with simian AIDS 28, 29, in peripheral blood mononuclear cell (PBMC) of Chinese–origin rhesus macaques to avoid immune response to cellular antigens of these 2 different sub-species of Rhesus monkeys…

2. Fig. 2B color coding is difficult to follow. Change groups A and B to different colors to make comparisons easier to follow.

3. Line 56 add reference number 14.

4. Line 404 … Animals were euthanized to avoid unnecessary suffering when reached humane endpoint, change to…when the experimental endpoint was reached. Humane treatment is guaranteed by animal committee approval from Foshan University and need not be repeated in this context.

PLOS authors have the option to publish the peer review history of their article (what does this mean?). If published, this will include your full peer review and any attached files.

Reviewer #1: No

Reviewer #2: **Yes: **Preston A. Marx

Figure Files:

Data Requirements:

Reproducibility:

References:

---

## [Decision Letter · Decision Letter 1]

15 May 2021

Dear Prof. Chen,

We are pleased to inform you that your manuscript 'Sustained viremia suppression by SHIVSF162P3CN-recalled effector-memory CD8+ T cells after PD1-based vaccination' has been provisionally accepted for publication in PLOS Pathogens.

Best regards,

Daniel C. Douek

Associate Editor

PLOS Pathogens

Richard Koup

Section Editor

PLOS Pathogens

Kasturi Haldar

Editor-in-Chief

PLOS Pathogens

orcid.org/0000-0001-5065-158X

Michael Malim

Editor-in-Chief

PLOS Pathogens

orcid.org/0000-0002-7699-2064

Reviewer Comments (if any, and for reference):

Reviewer's Responses to Questions

**Part I - Summary**

Reviewer #1: I am satisfied with author's responds to my comments

Reviewer #2: All questions satisfactorily addressed

**Part II – Major Issues: Key Experiments Required for Acceptance**

Reviewer #1: (No Response)

Reviewer #2: All questions satisfactorily addressed

**Part III – Minor Issues: Editorial and Data Presentation Modifications**

Reviewer #1: (No Response)

Reviewer #2: All questions satisfactorily addressed

PLOS authors have the option to publish the peer review history of their article (what does this mean?). If published, this will include your full peer review and any attached files.

Reviewer #1: No

Reviewer #2: **Yes: **Preston A Marx

---

## [Editor Report · Acceptance letter]

1 Jun 2021

Dear Prof. Chen,

We are delighted to inform you that your manuscript, " Sustained viremia suppression by SHIVSF162P3CN-recalled effector-memory CD8+ T cells after PD1-based vaccination
," has been formally accepted for publication in PLOS Pathogens.

Best regards,

Kasturi Haldar

Editor-in-Chief

PLOS Pathogens

orcid.org/0000-0001-5065-158X

Michael Malim

Editor-in-Chief

PLOS Pathogens

orcid.org/0000-0002-7699-2064